# Effect of Types of Dementia Care on Quality of Life and Mental Health Factors in Caregivers of Patients with Dementia: A Cross-Sectional Study

**DOI:** 10.3390/healthcare11091245

**Published:** 2023-04-27

**Authors:** Seung-Hyun Cho, Hyun-Se Choi

**Affiliations:** 1Department of Occupational Therapy, College of Health Sciences and Social Welfare, Woosuk University, Wanju 55338, Republic of Korea; kaic21@gmail.com; 2Department of Rehabilitation Medicine, Seoul National University Bundang Hospital, Seongnam 13620, Republic of Korea

**Keywords:** dementia, quality of life, depression, happiness, diagnostic self-evaluation

## Abstract

In Eastern cultures, particularly in South Korea, caregiving for dementia patients at home is common, yet even after facility placement, families may experience ongoing burden due to cultural factors. The aim of this study was to examine the burden experienced by caregivers of dementia patients, considering cultural factors influencing in-home care and facility-based care. Using a cross-sectional study design, we compared the quality of life, depression, subjective happiness, and subjective health of family caregivers providing in-home care (FCHC) and informal family caregivers (IFCGs). Data from the 2019 Community Health Survey conducted by the Korea Disease Control and Prevention Agency (KDCA) that met the study criteria were selected and statistically analyzed. The results showed that psychological/emotional and economic burdens were the primary burden factors for both FCHC and IFCGs. Statistically significant differences were found between the two groups in terms of quality of life, depression, subjective happiness, and subjective health. Specifically, FCHC demonstrated a lower quality of life, and both groups experienced moderate to severe depression, indicating the need for mental health management for caregivers of individuals with dementia. As not all FCHC can be transitioned to IFCGs, interventions tailored to specific caregiving types should be developed to improve the quality of life, depression, subjective happiness, and subjective health of caregivers of individuals with dementia.

## 1. Introduction

Dementia is a chronic or progressive syndrome caused by various brain diseases and is characterized by a decline in memory, cognition, and the ability to perform activities of daily living (ADLs) [1]. The number of patients with dementia worldwide is estimated to be approximately 55 million, with nearly 10 million new cases reported each year [2]. South Korea has the fastest-aging population in the world, with a rapidly increasing number of patients with dementia [3]. As of 2020, the prevalence of dementia in South Korea among patients aged 65 and older was 10.25%, and it is expected to increase to 15.91% by 2050, affecting approximately 3 million people [4]. Caregivers of patients with dementia account for the largest proportion of all patient caregivers and are required to spend more time providing care for several years [5]. As of 2020, the average cost of managing a single patient with dementia in South Korea was approximately KRW 20.6 million. While direct medical expenses are covered by the National Health Insurance Service, most non-medical expenses, such as caregiving and transportation, are borne by individuals without support [6]. Thus, the burden on caregivers of patients with dementia is substantial, and they may even be considered as potential secondary patients due to exposure to various risks such as cardiovascular diseases, chronic inflammation resulting from hypertension, and hyperactivity of the sympathetic nervous system [7,8]. Moreover, the heavy burden of care occasionally leads to suicidal thoughts in these caregivers [9]. Therefore, providing support to dementia caregivers is essential [10]. In particular, results of a previous study indicated that patients with dementia can stay at home for a longer duration if in-home caregivers receive appropriate and customized interventions [11]. The factors that contribute to the burden experienced by dementia caregivers are diverse, including quality of life, depression, subjective health, and subjective happiness [12,13,14]. Further, decreased quality of life among caregivers leads to poor quality of care provided to patients with dementia, thereby negatively affecting their progress [15]. Depression is a prevalent issue among dementia caregivers, with more than 50% of them reporting experiencing depression due to various psychological factors [16]. Caregiver depression has negative effects on both the caregiver and the patient, such as rapid cognitive decline [14]. The more negative the caregiver’s subjective health, the higher the caregiving burden and the more severe the patient’s symptoms become. Therefore, caregiver subjective health is an important factor to consider in order to reduce caregiver burden [17].

Dementia care can be divided into two categories: family caregivers providing in-home care (FCHC) and informal family caregivers (IFCGs). FCHC refers to care primarily provided by non-professionals, such as family members and friends, who offer most of the assistance and supervision required to meet the basic needs of dementia patients living in the community [18]. Meanwhile, IFCGs maintain an indirect caregiving role while the patient with dementia is under the care of professionals in a long-term care facility. In Eastern cultures such as South Korea, family care is generally preferred, with family members providing care for the patient in their own homes (FCHC). However, when the patient’s condition worsens, the type of care may shift to facility care, transitioning family members into the role of IFCGs. Even in the IFCG category, where professional caregivers provide direct care in long-term care facilities, family members may still maintain an indirect caregiving role and experience ongoing burden, such as guilt or a sense of responsibility, due to cultural factors. IFCG care involves care provided by professional personnel, such as home care nurses working in facilities such as group and nursing homes, where approximately 40% of dementia patients reside [19,20]. These professionals mainly provide care services for basic ADLs, such as meals, bathroom assistance, and communication with patients’ families, playing an important role in improving patients’ quality of life [21]. Particularly, family caregivers, especially those providing in-home care (FCHC), may experience increased burden due to their lack of professional knowledge and limited caregiving education [22]. For dementia care, the importance of both family-centered care, which includes informal caregivers such as family members, and community-centered care, which involves professional caregivers, is emphasized in the relevant literature. In the case of dementia patients, the role of decision making by caregivers in the community is particularly important due to many unexpected situations [23]. This highlights the need for adequate support and resources for both FCHC and IFCGs to effectively manage the complex challenges associated with dementia care. Wolff et al. [24] reported that both family and community care play a key role as cognitive function gradually deteriorates. Currently, community family care services are preferred over nursing facility services [25]. Ydstebø et al. [26] reported that expanded social networks, professional care, and the volume of provided care were reduced among patients receiving family care. In Korea, approximately 60% of patients with dementia in the communities receive care from their families, indicating that family support is crucial for older adults with dementia [4]. In a previous study on the quality of life and mental health among dementia caregivers, Park and Kim [27] reported that when caregivers are family members, their quality of life decreases, and their mental health is negatively affected. However, there is a lack of studies comparing the types of dementia care. This cross-sectional study was based on the hypothesis that the reasons for dementia burden would vary depending on the type of care provided to dementia patients. Specifically, we aimed to investigate the quality of life and mental health factors of caregivers of dementia patients, considering the type of caregiving they engage in. The ultimate goal of this study was to provide fundamental data that could contribute to the development of evidence-based intervention programs for caregivers of dementia patients, which would ultimately help to alleviate the burden associated with dementia care.

## 2. Materials and Methods

### 2.1. Design

This study employed a cross-sectional design to investigate the relationship between the variables of interest in family caregivers providing in-home care for dementia patients (FCHC) and informal family caregivers (IFCGs). The cross-sectional design was chosen because it allows for the collection and analysis of data from a large, diverse sample of caregivers at a single point in time, offering a snapshot of the current state of the caregiver population. This design is suitable for addressing the research questions and objectives as it facilitates the identification of potential correlations and patterns in the data, which can inform future longitudinal or experimental studies. Cross-sectional studies have several advantages, such as being relatively cost-effective, efficient, and easy to conduct compared to longitudinal designs. They are also useful for generating hypotheses and identifying potential associations between variables that can be further explored in future studies. However, cross-sectional studies have some limitations, including the inability to establish causality or temporal relationships between variables. Despite these limitations, the cross-sectional design was deemed appropriate for the present study as it provides valuable insights into the caregiving experiences and challenges faced by FCHC and IFCGs in the context of dementia care.

### 2.2. Data Collection

This study utilized data from the 2019 Community Health Survey [28] conducted by the Korea Disease Control and Prevention Agency (KDCA), which publicly provides the original survey data. The authors conducted research using this data. KDCA conducts a sample survey of adults aged ≥19 years, covering topics such as health condition, use of healthcare services, diseases, vaccinations and medical examinations, health knowledge, accidents and substance addiction, restriction on activities and quality of life, use of healthcare centers, socio-physical environment, educational and economic activities, and personal hygiene in accordance with the Regional Public Health Act. The 2019 KDCA Community Health Survey covered the entire country of South Korea, including 255 regions, with approximately 900 participants per region, resulting in a total of 228,610 respondents. The survey was conducted through face-to-face interviews by trained surveyors on a one-to-one basis between 16 August 2019, and 31 October 2019. This information provides context regarding the scope and methodology of the survey used in the present study.

### 2.3. Participants

The present study extracted data from the KDCA’s Community Health Survey (2019) that met the criteria established by the researchers. Specifically, the study focused on two types of caregivers: (1) family caregivers providing in-home care (FCHC), who primarily provide care for family members with dementia in the home setting, and (2) informal family caregivers (IFCGs), who are family members of dementia patients that have been admitted to long-term care facilities but still maintain a caregiving role due to cultural factors. This classification accounts for the cultural and systemic preference in South Korea for families to care for dementia patients at home, while also acknowledging the circumstances that may necessitate the use of long-term care facilities. A total of 7133 cases of data from family members of dementia patients in Korea who responded to the Community Health Survey were collected for this study.

#### 2.3.1. Family Caregivers Providing In-Home Care of Patients with Dementia (FCHC)

Family caregivers providing in-home care of patients with dementia are defined by the following criteria: (1) caregivers having a family member diagnosed with dementia, (2) caregivers living together with the patient diagnosed with dementia, and (3) caregivers mostly caring for the patient with dementia at home. FCHC play a significant and direct role in providing care to dementia patients within the family home. Overall, 2630 participants were classified as FCHC.

#### 2.3.2. Informal Family Caregivers (IFCGs)

Informal family caregivers are defined by the following criteria: (1) family members having a relative diagnosed with dementia, (2) family members not living together with the patient diagnosed with dementia, and (3) family members maintaining an indirect caregiving role while the patient with dementia is under the care of professionals in a long-term care facility. IFCGs may experience ongoing burden, such as guilt or a sense of responsibility, due to cultural factors even though the patient is placed in a care facility. Overall, 4503 participants were classified as IFCGs.

We utilized all data classified into the FCHC and IFCG groups and did not use any specific sampling strategies. Missing data were treated as missing values and excluded from the analysis. The FCHC group consisted of 2583 cases after excluding 47 cases out of 2630, while the IFCG group consisted of 4483 cases after excluding 20 cases out of 4503.

### 2.4. Variables

In this study, quality of life was assessed using the EQ-5D-3L variable, while mental health factors, including depression, subjective happiness, and subjective health, were also analyzed. Notably, all of these variables, encompassing the EQ-5D-3L, are well-established and reliable measurement tools widely employed in the development and implementation of public health policies in Korea [29].

#### 2.4.1. Quality of Life

Quality of life was assessed using the three-level version of the EuroQol-5 dimension (EQ-5D-3L) [30], which measures mobility, self-care, usual activities, pain/discomfort, and anxiety/depression. The EQ-5D-3L consists of five dimensions, each with three response levels: “no problems”, “some problems”, and “extreme problems”. The EQ-5D index is an indicator that integrates technology systems of five levels of quality of life related to health, with scores ranging from −0.0171 to 1. Lower scores indicate a poorer quality of life. The instrument’s validity has been established in previous research, and its Cronbach’s alpha value of 0.794 demonstrates a reliable level of internal consistency. The Community Health Survey [28] of KDCA provided the following formula for calculating the weight values used to obtain the EQ-5D index:EQ-5D index = 1 − (0.05 + 0.096(M2) + 0.418(M3) + 0.046(SC2) + 0.136(SC3) + 0.051(UA2) + 0.208(UA3) + 0.037(PD2) + 0.151(PD3) + 0.043(AD2) + 0.158(AD3) + 0.05(N3))

M2, M3: mobility; SC2, SC3: self-care; UA2, UA3: usual activities; PD2, PD3: pain/discomfort; AD2, AD3: anxiety/depression; and N3: extreme problems other than those measured in the abovementioned five areas.

#### 2.4.2. Depression

Depression was assessed using the Patient Health Questionnaire-9 (PHQ-9), a nine-item instrument designed to diagnose major depressive disorder. Items are scored on a four-point scale, with higher scores indicating higher levels of depression [28,31]. The PHQ-9 has been validated in numerous studies and exhibits a Cronbach’s alpha value of 0.800, indicating a reliable level of internal consistency.

#### 2.4.3. Subjective Happiness

Subjective happiness was measured using a single item asking participants to rate their overall satisfaction with life on a scale from 1 (very dissatisfied) to 10 (very satisfied). This single-item measure has been used in various studies to assess subjective happiness and has demonstrated validity in these contexts [28,32].

#### 2.4.4. Subjective Health

Subjective health was assessed using a single item asking participants to rate their overall health. Responses were scored on a five-point scale, with higher scores indicating better subjective health after reverse coding for statistical analysis. This single-item measure has been validated in previous research for assessing subjective health [28,33].

### 2.5. Statistical Analysis

We set the significance level at *p* < 0.05 for our statistical analysis. Descriptive statistics and frequencies were employed to analyze the general characteristics of the study participants. To compare the data between the FCHC and IFCG groups, we performed independent sample t-tests using only continuous variables. Additionally, we conducted multiple regression analyses using only continuous variables to determine the effects of independent variables on quality of life, depression, subjective happiness, and subjective health. Since all regression models were subjected to the Durbin–Watson test to check for the presence of autocorrelation in the residuals and the multicollinearity test to detect any multicollinearity issues, and no such issues were observed, it can be concluded with confidence that the results of the regression analyses are reliable. The assessments utilized for the FCHC and IFCG groups were identical, ensuring comparability between the two groups. We considered only the common characteristics of caregivers for dementia patients and the characteristics used to classify participants into the FCHC and IFCG groups, assuming that general characteristics such as age and gender did not significantly impact the results. We did not perform any subgroup analyses or examine interactions.

### 2.6. Ethical Statement

This study was conducted in accordance with the principles of the Declaration of Helsinki and was approved for exemption from review by the Institutional Review Board of Woosuk University (WS-2023-01). The Community Health Survey was approved for use by the Institutional Bioethics Committee of KDCA until 2016. However, since 2017, it has been exempted from review because it did not meet the criteria of human subject research in accordance with the Enforcement Rule of Bioethics and Safety Act. The data used in the study was de-identified and publicly accessible, rendering it eligible for exemption from institutional review board review.

## 3. Results

### 3.1. General Characteristics of the Study Populations

The FCHC and IFCG groups showed differences in several characteristics. In both groups, women constituted the majority of caregivers. The mean age of patients in the FCHC group (63.52 years) was higher than that in the IFCG group (52.97 years). The FCHC group had a lower average monthly household income (USD 2137) compared to the IFCG group (USD 3074). The majority of caregivers in the FCHC group were parents (62.3%), while in the IFCG group, an even higher percentage of caregivers were parents (87.6%). The primary reason for the burden of dementia care in the FCHC group was psychological and emotional (51.6%), while in the IFCG group, it was more evenly distributed between psychological and emotional (41.2%) and financial (33.8%) factors. The participation rates in social activities varied between the groups, with the IFCG group generally showing higher participation rates across different types of activities. Detailed information on the general characteristics of the study populations can be found in Table 1.

### 3.2. Comparison between FCHC and IFCGs

The difference in the quality of life between the FCHC (M = 0.835; SD = 0.186) and IFCG (M = 0.908; SD = 0.088) groups was statistically significant (*t* = −18.92; *p* < 0.001). Further, the difference in the depression rates between the FCHC (M = 12.18; SD = 4.27) and IFCG (M = 11.29; SD = 3.15) groups was also statistically significant (*t* = 9.33; *p* < 0.001). Moreover, the difference in subjective happiness was statistically significant between the FCHC (M = 6.36; SD = 1.91) and IFCG (M = 6.94; SD = 1.74) groups (*t* = −12.76; *p* < 0.001). Furthermore, the difference in subjective health was also statistically significant between the FCHC (M = 2.82; SD = 1.00) and IFCG (M = 3.16; SD = 0.85) groups (*t* = −14.98; *p* < 0.001) (Table 2).

### 3.3. Factors Affecting Quality of Life

The factors that significantly affected the quality of life were depression, subjective happiness, and subjective health (*p* < 0.001). Particularly, in the FCHC group, for every one-point increment in the depression score, the quality of life decreased by 0.013, and for every one-point increment in subjective happiness, the quality of life increased by 0.007. Moreover, for every one-point increment in subjective health, the quality of life increased by 0.068.

In the IFCG group, for every one-level increment in the depression score, the quality of life decreased by 0.007, and for every one-point increment in subjective happiness and subjective health, the quality of life increased by 0.005 and 0.033, respectively.

In both FCHC and IFCGs, quality of life was mostly affected by subjective health, followed by depression and subjective happiness (Table 3).

### 3.4. Factors Affecting Depression

The factors that significantly affected depression were quality of life, subjective happiness, and subjective health (*p* < 0.001). In particular, in the FCHC group, for every one-point increment in the quality of life, the depression score decreased by 7.368. Further, for every one-point increment in subjective happiness and subjective health, the depression score decreased by 0.603 and 0.631, respectively. Furthermore, depression was mainly affected by the quality of life, followed by subjective happiness and subjective health.

In the IFCG group, for every one-point increment in the quality of life, the depression score decreased by 9.265. For every one-point increment in subjective happiness and subjective health, the depression score decreased by 0.502 and 0.5, respectively. Furthermore, depression was mainly affected by subjective happiness, followed by quality of life and subjective health (Table 4).

### 3.5. Factors Affecting Subjective Happiness

The factors that significantly affected subjective happiness were quality of life, depression, and subjective health (*p* < 0.001). Particularly, in the FCHC group, for every one-point increment in the quality-of-life score, subjective happiness increased by 0.941. For every one-point increment in the depression score, subjective happiness decreased by 0.136. In addition, for every one-point increment in subjective health, subjective happiness increased by 0.394.

In the IFCG group, for every one-point increment in the quality-of-life score, subjective happiness increased by 1.971. For every one-point increment in the depression score, subjective happiness decreased by 0.162. Additionally, for every one-point increment in subjective health, subjective happiness increased by 0.405.

In both the FCHC and IFCG groups, subjective happiness was mainly affected by depression, followed by subjective health and quality of life (Table 5).

### 3.6. Factors Affecting Subjective Health

The factors that significantly affected subjective health were quality of life, depression, and subjective happiness (*p* < 0.001). Particularly, in the FCHC group, for every one-point increment in the quality-of-life score, subjective health increased by 2.141, and for every one-point increment in the depression score, subjective health decreased by 0.034. In addition, for every one-point increment in subjective happiness, subjective health increased by 0.094.

In the IFCG group, for every one-point increment in the quality-of-life score, subjective health increased by 3.153, and for every one-point increment in depression score, subjective health decreased by 0.037. In addition, for every one-point increment in subjective happiness, subjective health increased by 0.092.

In both the FCHC and IFCG groups, subjective health was mainly affected by quality of life, followed by subjective happiness and depression (Table 6).

## 4. Discussion

This study examined the quality of life, depression level, subjective happiness, and subjective health of FCHC and IFCGs, Moreover, we evaluated the characteristics of burden related to dementia care among caregivers stratified by the type of care provided and the characteristics of their economic/social activities. In a previous study comparing general caregivers not involved in caregiving activities with FCHC, FCHC were at a higher risk of presenting with worse health status, weight loss, anxiety, depression, and poor quality of life [34]. In other words, caregivers exhibit poor physical and mental health owing to the caregiver burden. We found that both FCHC and IFCGs experience severe depression, suggesting the necessity to manage the mental health of dementia caregivers. In particular, based on the results of PHQ-9—the instrument used in this study to measure depression—11 was the best score in terms of sensitivity and specificity in the optimal cut-off score study for diagnosing depression, which is usually ~1 point below the standard, suggesting 10 points [35]. The average PHQ-9 scores of 12.18 and 11.29 among FCHC and IFCGs, respectively, indicate a high level of depression among dementia caregivers. This finding is consistent with previous research, which found that 58.7% of dementia caregivers exhibit symptoms of depression, and 21.7% have severe depression [36].

Compared with the IFCG group, the FCHC group demonstrated negative results for all variables. These results are consistent with previous studies that reported that FCHC burden and feelings of isolation are characterized by an increasing trend [37]. Notably, FCHC experience extreme pressure, resulting in the development of physical/psychological problems [38].

Moreover, a previous study investigating the baseline quality of life among dementia caregivers reported that the average scores of quality of life of caregivers for patients with mild, moderate, and severe dementia were 0.86 (SD = 0.18), 0.85 (SD = 0.19), and 0.82 (SD = 0.23), respectively [39]. In the present study, the average score of quality of life in FCHC was 0.83, and based on the results of previous studies, this score indicates the condition of the quality of life of caregivers for patients with mild-to-moderate dementia. In contrast, the average quality of life score in IFCGs was 0.908, which is higher than the score of caregivers for patients with mild dementia (0.86) in a previous study. Moreover, a previous study [40] reported that the quality of life is a key factor in deciding to transition from an FCHC to an IFCG type of care; when an FCHC has a decreased quality of life, they shift their role to that of an IFCG, resulting in a relatively improved quality of life compared to when they were an FCHC. This result indicates that facility care is more beneficial than family care in terms of the quality of life of caregivers.

Notably, quality of life affects subjective health, which in turn affects quality of life. Park et al. [41] reported that quality of life is closely related to subjective health; thus, if the quality of life declines, the subjective health level also decreases. A previous study reported that caregivers’ subjective health is positively correlated with their quality of life, and as their health status improves, their quality of life improves [42]. Moreover, the health status of dementia caregivers is consistently related to their quality of life, and good subjective health status leads to an improvement in the quality of life [43].

It is necessary to improve subjective health, depression, and subjective happiness to improve the quality of life among dementia caregivers, and there was no difference between the FCHC and IFCG groups in these factors in our study. In particular, our study found that subjective health and depression greatly affect the caregivers’ quality of life. A previous study on the predictive factors for the quality of life among dementia caregivers reported that caregivers’ depression is the strongest predictive factor. Thus, caregivers taking care of patients with physical, mental, emotional, financial, and social burdens demonstrate an increased possibility of exhaustion, worsened mental health, and decreased quality of life [44]. Moreover, dementia caregivers exhibited a high level of anxiety, depression, and burden in addition to other psychological symptoms, such as obsession, hostility, and somatization, which considerably affect their quality of life [45]. Depression has been reported to affect their quality of life and subjective health [46], and subjective happiness has been known to be remarkably related to depression. Furthermore, at higher levels of depression, the caregivers’ happiness decreases [47]. Notably, it has been reported that depression is a predictive factor of quality of life among family caregivers [48]. In addition, a previous study reported that depression affects subjective happiness [49]. Moreover, higher levels of depression decrease subjective happiness in caregivers [50]. In Spain, a study on subjective happiness and depression reported that subjective happiness is negatively correlated with depression; thus, with increased subjective happiness, the depression level decreases [51]. It is essential to ensure that the dementia caregivers’ subjective health is maintained because it positively affects patients and indicates improved quality of life among the caregivers. A previous study on caregivers’ exhaustion, specifically emotional exhaustion, revealed that quality of life is significantly related to depression and low level of subjective health [52]. Moreover, it is important for the caregivers to participate in activities providing pleasure and a sense of accomplishment while maintaining social relationships [53]. Notably, dementia caregivers exhibit symptoms of being more tired and overwhelmed in a state of social isolation [54].

FCHC and IFCGs should receive appropriate interventions. Individualized occupational therapy can effectively alleviate their burden, leading to an improvement in their quality of life, frequency of engagement in activities, and capability for self-care [55]. These interventions for dementia caregivers increased their control of their lives and improved their mood and health status, as confirmed during 12 weeks of follow-up in a previous study [56]. In particular, it is considered that the use of these interventions can most effectively improve the caregivers’ quality of life, depression level, subjective health, and subjective happiness [57].

Professional interventions should be conducted to decide the optimal timing for converting FCHC to IFCGs, which need to be considered when developing interventions in the future [58]. Caregiver burden reportedly decreases by 26.1% in the cases in which patients are temporarily admitted to short-term care facilities [59]. Owing to the limited healthcare resources, there will be a limitation in transferring all patients with dementia to long-term care facilities. Therefore, as a method for reducing FCHC’s burden, temporary placement of patients at short-term care facilities can be considered.

Based on our findings, IFCGs are more likely to be involved in economic and social activities compared with FCHC, probably because they are relatively more relieved from caregiving burden. Previous studies comparing FCHC and IFCG reported that the emotional and physical problems of caregivers affected their ability to take care of patients with dementia, leading to earlier facility admission; thus, it is important to satisfy the needs of FCHC. Additionally, support for IFCGs is as important as support for FCHC [60,61,62]. In a qualitative study on the needs of caregivers who transitioned from being an FCHC to an IFCG, caregivers most frequently reported their emotional concerns, knowledge, information, and need for support; further, the study reported that both FCHC and IFCGs needed more information on guidance [63]. We believe that these results support our study findings, recommending that measures should be provided according to the characteristics of care provided by caregivers because the quality of life, depression level, subjective happiness, and subjective health differ between FCHC and IFCGs.

Our study has some limitations. First, the type of dementia in patients cared for by their caregivers and their functions were not determined, although dementia caregivers’ quality of life, depression, subjective happiness, and subjective health were investigated. Regarding the type of dementia, such as Alzheimer’s disease or vascular dementia, there may be differences in caregivers’ experience. In addition, when dementia is mild, instrumental ADL decreases, whereas when it is moderate, basic ADL decreases. Thus, caregivers may have different experiences according to the characteristics of their patients [64]. Therefore, it is suggested to classify caregivers by type and function of patients with dementia in future studies. Second, our study revealed poor mental health of caregivers of patients with dementia, but it did not reveal the possible risk of cognitive decline [65] caused by poor mental health. Therefore, further studies are required to focus on the caregivers’ cognitive abilities. Third, although the quality of life can be evaluated according to various areas [66], our study only investigated the quality of life measured using EQ-5D, which measures mobility, self-care, usual activities, pain/discomfort, and anxiety/depression. Thus, in the future, studies are required to evaluate various variables related to quality of life, such as financial ability. Fourth, this study reflects the oriental culture of South Korea, where family care for people with dementia is prioritized, and when the patient’s condition worsens, they are institutionalized. Due to the cultural characteristics of Korea, the burden of caregiving may still remain even when the family transitions to institutionalized care. These characteristics may be different in other countries or cultures, so the generalizability of this study’s findings may be limited. Fifth, while this study assumes that general characteristics such as age and gender have no effect, we suggest that future studies consider methods such as PS matching to control for confounding in the paper.

## 5. Conclusions

This study classified caregivers of dementia patients into two groups: family caregivers providing in-home care (FCHC), who primarily provide care for family members with dementia in the home setting, and informal family caregivers (IFCGs), who are family members of dementia patients admitted to long-term care facilities for various reasons. The study investigated the characteristics of these two groups concerning quality of life, depression, subjective well-being, subjective health, and dementia-related burden. The results showed that FCHC had a lower quality of life and more negative mental health outcomes than IFCGs. Both groups exhibited moderate to severe depression, underscoring the need for mental health care for caregivers of people with dementia. FCHC showed poorer quality of life, indicating that quality of life is a key determinant of care transitions. However, given the practical constraints, it may not be feasible to convert all FCHC to IFCGs. Therefore, interventions for caregivers of people with dementia should be developed, considering the characteristics of both FCHC and IFCGs, to improve their quality of life, depression, subjective well-being, and subjective health.

## Figures and Tables

**Table 1 healthcare-11-01245-t001:** General characteristics of the study populations.

	FCHC		IFCGs	
	N	%	N	%
Gender				
Men	1172	44.6	2093	46.5
Women	1458	55.4	2410	53.5
Housing type				
General houses	1914	72.8	2410	53.5
Apartments	716	27.2	2093	46.5
Caregiver–dementia patient relationships				
Parents	1638	62.3	3945	87.6
Spouse	715	27.2	214	4.8
Siblings	42	1.6	323	7.2
Patient themselves	220	8.4	10	0.2
Children	14	0.5	4	0.1
Determinants of dementia care burden				
Financial	460	17.5	1522	33.8
Psychological and emotional	138	51.6	1853	41.2
Physical	309	11.7	149	3.3
Related to time	166	6.3	233	5.2
Social prejudice	9	0.3	31	0.7
No burden	318	12.1	697	15.5
Economic activity				
Yes	1282	48.7	2971	66.0
No	1343	51.1	1530	34.0
Social activity (Yes)				
Religious	777	29.5	1364	30.3
Amity	1220	46.4	2760	61.3
Leisure	499	19.0	1524	33.8
Voluntary	177	6.7	497	11.0
	**M**	**SD**	**M**	**SD**
Age	63.52	17.42	52.97	15.20
Monthly income (USD)	2137.59	1953.08	3074.35	2197.31

**Table 2 healthcare-11-01245-t002:** Comparison between FCHC and IFCGs.

Variables	FCHC		IFCGs		*t*	*p*
	M	SD	M	SD		
Quality of life	0.835	0.186	0.908	0.088	−18.92	0.000 ***
Depression	12.18	4.27	11.29	3.15	9.33	0.000 ***
Subjective happiness	6.36	1.91	6.94	1.74	−12.76	0.000 ***
Subjective health	2.82	1.00	3.16	0.85	−14.98	0.000 ***

*** *p* < 0.001.

**Table 3 healthcare-11-01245-t003:** Factors affecting the quality of life.

Independent Variables		FCHC						IFCGs				
	*B*	*β*	*t*	*95% CI*		*VIF*	*B*	*β*	*t*	*95% CI*		*VIF*
				*LL*	*UL*					*LL*	*UL*	
Depression	−0.013	−0.297	−16.256 ***	−0.014	−0.011	1.351	−0.007	−0.255	−17.746 ***	−0.008	−0.006	1.264
Subjective happiness	0.007	0.075	4.171 ***	0.004	0.010	1.308	0.005	0.093	6.448 ***	0.003	0.006	1.264
Subjective health	0.068	0.373	21.000 ***	0.062	0.075	1.281	0.033	0.318	22.726 ***	0.030	0.036	1.201
		Durbin–Watson = 1.921 R^2^ = 0.364, Adj R^2^ = 0.364, F = 492.991, *p* < 0.001						Durbin–Watson = 1.876 R^2^ = 0.269, Adj R^2^ = 0.269, F = 550.276, *p* < 0.001				

*** *p* < 0.001; *VIF* = variance expansion factor.

**Table 4 healthcare-11-01245-t004:** Factors affecting depression.

Independent Variables		FCHC						IFCGs				
	*B*	*β*	*t*	*95% CI*		*VIF*	*B*	*β*	*t*	*95% CI*		*VIF*
				*LL*	*UL*					*LL*	*UL*	
Quality of life	−7.368	−0.313	−16.256 ***	−8.256	−6.479	1.427	−9.265	−0.258	−17.746 ***	−10.289	−8.241	1.279
Subjective happiness	−0.603	−0.270	−15.206 ***	−0.681	−0.525	1.208	−0.502	−0.277	−19.946 ***	−0.552	−0.453	1.172
Subjective health	−0.631	−0.147	−7.511 ***	−0.796	−0.466	1.468	−0.500	−0.135	−9.154 ***	−0.607	−0.393	1.315
		Durbin–Watson = 1.812 R^2^ = 0.329, Adj R^2^ = 0.328, F = 420.920, *p* < 0.001						Durbin–Watson = 1.783 R^2^ = 0.261, Adj R^2^ = 0.260, F = 527.147, *p* < 0.001				

*** *p* < 0.001; *VIF* = variance expansion factor.

**Table 5 healthcare-11-01245-t005:** Factors affecting subjective happiness.

Independent Variables		FCHC						IFCGs				
	*B*	*β*	*t*	*95% CI*		*VIF*	*B*	*β*	*t*	*95% CI*		*VIF*
				*LL*	*UL*					*LL*	*UL*	
Quality of life	0.941	0.089	4.171 ***	0.499	1.383	1.563	1.971	0.099	6.448 ***	1.372	2.571	1.356
Depression	−0.136	−0.305	−15.206 ***	−0.154	−0.119	1.367	−0.162	−0.294	−19.946 ***	−0.178	−0.146	1.243
Subjective health	0.394	0.205	9.937 ***	0.316	0.472	1.444	0.405	0.198	13.187 ***	0.345	0.466	1.289
		Durbin–Watson = 1.752 R^2^ = 0.240, Adj R^2^ = 0.239, F = 271.994, *p* < 0.001						Durbin–Watson = 1.751 R^2^ = 0.216, Adj R^2^ = 0.216, F = 411.674, *p* < 0.001				

*** *p* < 0.001; *VIF* = variance expansion factor.

**Table 6 healthcare-11-01245-t006:** Factors affecting subjective health.

Independent Variables		FCHC						IFCGs				
	*B*	*β*	*t*	*95% CI*		*VIF*	*B*	*β*	*t*	*95% CI*		*VIF*
				*LL*	*UL*					*LL*	*UL*	
Quality of life	2.141	0.391	21.000 ***	1.941	2.341	1.344	3.153	0.325	22.726 ***	2.881	3.425	1.227
Depression	−0.034	−0.146	−7.511 ***	−0.043	−0.025	1.458	−0.037	−0.136	−9.154 ***	−0.045	−0.029	1.328
Subjective happiness	0.094	0.180	9.937 ***	0.075	0.112	1.268	0.092	0.189	13.187 ***	0.079	0.106	1.228
		Durbin–Watson = 1.863 R^2^ = 0.333, Adj R^2^ = 0.332, F = 429.550, *p* < 0.001						Durbin–Watson = 1.919 R^2^ = 0.253, Adj R^2^ = 0.253, F = 506.338, *p* < 0.001				

*** *p* < 0.001; *VIF* = variance expansion factor.

## Data Availability

The data are available from Korea Disease Control and Prevention Agency website (https://chs.kdca.go.kr/chs, accessed on 1 December 2022).

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
