# Peer review of "Effect of Types of Dementia Care on Quality of Life and Mental Health Factors in Caregivers of Patients with Dementia: A Cross-Sectional Study"

_healthcare, 2023, doi:10.3390/healthcare11091245_

Round 1

Reviewer 1 Report (New Reviewer)

Dear authors:

Keywords:

The keyword "Subjective Happines" is not a MeSH descriptor, the correct descriptor corresponds to "Happiness" for proper indexing of the manuscript. Likewise, the appropriate MeSH term for "Subjective health" is "Diagnostic Self Evaluation".

Abstract:

The summary should detail, to the extent possible, the methodology used to provide information on design, population (sample), instruments, data collection and analysis, etc.

Introduction:

In the introduction (page 2, line 61 and following): should be identified with the acronym Family Caregivers Providing In Home Care (FCHC) on first use only. In my opinion it is preferable to define these concepts of FCHC and IFCG before describing the characteristics of caregivers in the context studied.

Materials and Methods:

The structure of the method should have a logical order (design, population, sample, data collection, instruments and variables, application of the instruments, data analysis, ethical statement) starting with a subheading on design. Ethical aspects should be placed at the end of the method, separating data collection from this subheading.

Data on population, sample, inclusion and exclusion criteria should be correctly identified under a single subheading.  It would be appropiate to report how they have calculated the sample size estimatation.

They should explain in a more orderly manner the information of each of the instruments used: in addition to reporting the reliability with cronbach's alpha, they should include the validity (dimensions in which it is structured) and detail the number of items, response scale, score of the items, total scores of the instrument.

Results:

Do not duplicate in the text of the manuscript the results in Table 1, point out only the most relevant ones in the text.

Unify the font size in tables.

References:

Try to reduce as much as possible the use of references older than 10 years (n=23).

Author Response

Dear Peer Reviewers,

We would like to express our sincere gratitude for your valuable feedback and insightful comments on our manuscript. Your expertise and constructive suggestions have significantly contributed to improving the quality of our work. We appreciate the time and effort you have dedicated to reviewing our manuscript, and we believe that the revisions made in response to your comments have strengthened our study.

Keywords:
Based on your suggestion, we have revised 'Subjective Happiness' to 'Happiness' and 'Subjective Health' to 'Diagnostic Self Evaluation.'

Abstract:
We have made revisions to include more details on the methodology of our study, as you suggested. Specifically, we added a sentence to describe the data source and statistical analysis used in the study.

Introduction:
We have modified the concepts of FCHC and IFCG by defining them first in the introduction and then describing their associated properties.

Materials and Methods:
We rearranged the content in a logical order and separated the data collection and ethical aspects. 

Results:
We've only included the most relevant findings from the results in Table 1. Additionally, we have unified the font size in the tables.

References:
We have replaced references older than 10 years with more recent references wherever possible. However, in cases where the original article better describes the research, we have retained the older reference, as newer references are not necessarily superior.

This manuscript is a resubmission of an earlier submission. The following is a list of the peer review reports and author responses from that submission.

Round 1

Author Response

We would like to express our gratitude for the thoughtful review of our paper. Based on your valuable feedback, we have made revisions to address the issues raised.

We have made modifications to the use of English and sentence structure throughout the paper. We have also improved the wording for ambiguous expressions.

We believe that there was confusion regarding the terms "Home caregivers" and "Facility caregivers." Therefore, we have modified the expressions for the subjects we are researching in this paper, which are family members of dementia patients.

Specifically, we have changed the subjects to the following:

2.2.1. Family caregivers providing in-home care of patients with dementia (in-home caregivers;HCG)
2.2.2. Family members of long-term care facility patients with dementia (long-term caregivers;LCG) 

In Eastern cultures, particularly in Korea, it is common for family members to prioritize caring for their loved ones with dementia at home. However, due to circumstances such as the worsening condition of the patient, they may need to be placed in a care facility. As a result of this cultural characteristic, the burden of caregiving may still be felt by family members even after transitioning to facility care. We have added additional content related to this topic.

In Korea, efforts are being made to provide institutional support to all families with dementia patients, and relevant research is also being conducted. This study was conducted with the goal of comparing and analyzing the factors related to caregiving burden according to the type of care - HCG or LCG - and discussing ways to overcome problems related to dementia within the constraints of limited healthcare resources.

This study was conducted while taking into account the cultural characteristics of the Eastern culture and we hope that the value of this study, which is derived from these cultural characteristics, is considered.

We have undergone professional English language editing and are attaching the corresponding documentation as evidence thereof.

Reviewer 2 Report

The results of the study are relevant and innovative, but the author needs to clarify some aspects better.

Introduction:

It is described that caregivers spend time for care, it is suggested to also include financial resources.

The importance of family- and community-centered care is described, it would be interesting to cite studies on experiences of compassionate communities.

Methodology

In variables, it is suggested to include information on the validation study and adaptation of instruments for the research context.

Conclusion

The results are about effects on caregivers and seem to suggest that care in LCGs would be preferable to home care. Explain better that, considering the importance of family home care for patients, one should think of strategies to improve the quality of life and other subjective aspects of these caregivers.

Author Response

We would like to express our gratitude for the thoughtful review of our paper. Based on your valuable feedback, we have made revisions to address the issues raised.

Introduction:
In response to the reviewer's comments, we have added information on the consumption of other resources related to caregiving in the introduction. Additionally, we have included references to studies highlighting the importance of family and community-centered care.

Methodology:
We have addressed the reviewer's concerns regarding the validity and reliability of the variables used in the study by including information on the validation and adaptation of instruments for the research context.

Conclusion:
We would like to emphasize that the cultural and institutional context in Korea, where family members are typically the primary caregivers for dementia patients, is an important factor to consider when interpreting the results of this study. While the results suggest that care in LCGs may be preferable to home care, it is important to note that LCG care requires significant healthcare resources. This study has been conducted with a recognition of the reality of limited healthcare resources and the constraints that this places on caregiving options.

In conclusion, it is necessary to consider the unique characteristics of HCG and LCG care, including cultural factors and limitations on healthcare resources. In Korea, efforts are being made to provide institutional support for all dementia patients and their families, and related research is ongoing. This study was conducted with the aim of contributing to this ongoing effort.

Reviewer 3 Report

The paper describes a study that examined the characteristics, quality of life, level of depression, subjective well-being, subjective health, and burden associated with dementia in a family caregiver group (HCG) and a long-term caregiver group (LCG). The study used the Korean Disease Control and Prevention Agency (KDCA) Community Health Survey as a data source and various measurement tools to assess the outcomes of interest. However, the definitions of HCG and LCG in this paper are ambiguous, for example, for patients who only briefly utilized the facility, which was not reported in this study. For this paper some statistical analyses and regression coefficients for various factors influencing these outcomes are reported, but no specific adjustment information for confounding factors is provided. 

Line2 and line 10-24 The paper does not provide a clear indication of the study's design with a commonly used term in the title or the abstract.

Line 81-86        There are no specified hypotheses mentioned in the introduction.

Line98-103       The present study discloses the original data of the Community Health Survey. However, the paper does not provide information about the locations, relevant dates, periods of recruitment, exposure, or follow-up.

Line105-108      The paper does not provide specific information on the eligibility criteria for participants or the methods of participant selection.

Line152-157      The comparability of assessment methods between the two groups (In-home Caregiver and Long-term Caregiver) is not explicitly discussed in the paper

Line89-96        The paper did not provide information on how the study size was arrived at.

Line152-157      the paper does not explicitly mention any methods used to control for confounding. 

Line152-157      The paper did not mention any methods used to examine subgroups and interactions.

Line152-157      The paper did not provide information on how missing data were addressed.

Line152-157      The paper does not mention any specific analytical methods used to account for the sampling strategy

Line152-157      The paper does not explicitly mention any sensitivity analyses. 

Line160-170      The paper did not provide information on the number of participants with missing data 

Line160-170      There is no specific line number that reports numbers of outcome events or summary measures in the paper provided. The paper describes the measurement tools used to assess subjective happiness, subjective health, and quality of life, but it does not provide specific outcome data or summary measures.

Line158-251      The paper provided describes the measurement tools used to assess subjective happiness, subjective health, and quality of life, and reports statistical analyses and regression coefficients for various factors affecting these outcomes. However, it does not provide specific estimates with confidence intervals or information on which confounders were adjusted for, category boundaries when continuous variables were categorized, or estimates of relative or absolute risk. Additionally, the text does not contain information on other analyses done, such as analyses of subgroups and interactions, or sensitivity analyses.

Author Response

We would like to express our gratitude for the thoughtful review of our paper. Based on your valuable feedback, we have made revisions to address the issues raised.

First, we clarified the study design as a cross-sectional study in the abstract. Second, we added hypotheses to the introduction section to provide more information about the study.
Thrid, We acknowledge that due to cultural and institutional differences in Korea, we may not have accurately conveyed the definitions of HCG and LCG. As a result, we have made revisions to clarify these definitions.
2.2.1. Family caregivers providing in-home care of patients with dementia (in-home caregivers;HCG)
2.2.2. Family members of long-term care facility patients with dementia (long-term caregivers;LCG) 

2.1. Data and Ethical Approval
We provided detailed information about the original data source, including the location, relevant dates, and recruitment period. 

2.2. Participants
In the participants section, we mentioned that we only extracted data that met the predefined criteria, and we described these criteria in detail.

2.4. Statistical Analysis
In the statistical analysis section, we stated that we used the same evaluation method in both groups and that no specific method was used to control for confounding. However, we assumed that general characteristics, such as age and gender, did not affect the study's outcomes. In addition, we suggested using methods such as PS matching to control for confounding in future research. We also classified participants into two groups (HCG and LCG) for comparison analysis and did not investigate subgroups or interactions. We handled missing data by excluding them from the analysis, and we provided specific information about the number of participants with missing data.

Results
In the Results section, we provided descriptive statistics, such as the means, in Table 2. In addition, we reported the results of regression analyses in Tables 3-6 and specified the 95% confidence intervals. Furthermore, we provided information related to the reliability of the analysis results.

Round 2

Reviewer 1 Report

This manuscript is much improved and easier to understand. There is no need to repeat the quality measures for the regression analysis with each variable described. One statement will suffice. 

Author Response

We express our appreciation for your thoughtful review of our manuscript. Based on your insightful feedback, we have made revisions to address the raised concerns.

To avoid repetition of the mention of reliability in each regression analysis result, we have made changes to the manuscript. In the statistical analysis section, we now specify that we conducted relevant tests to confirm the absence of autocorrelation in the residuals and multicollinearity issues. We have also made it clear that these tests were carried out to address the reliability of the regression analysis results.

Thank you for your valuable input, which helped improve the quality of our manuscript.